# T1a Glottic Cancer: Advances in Vocal Outcome Assessment after Transoral CO_2_-Laser Microsurgery Using the VEM

**DOI:** 10.3390/jcm10061250

**Published:** 2021-03-17

**Authors:** Wen Song, Felix Caffier, Tadeus Nawka, Tatiana Ermakova, Alexios Martin, Dirk Mürbe, Philipp P. Caffier

**Affiliations:** 1Department of Audiology and Phoniatrics, Charité–Universitätsmedizin Berlin, Corporate Member of Freie Universität Berlin and Humboldt-Universität zu Berlin, Campus Charité Mitte, Charitéplatz 1, D-10117 Berlin, Germany; wensonwen@hotmail.com (W.S.); felix.caffier@charite.de (F.C.); tadeus.nawka@charite.de (T.N.); dirk.muerbe@charite.de (D.M.); 2Fraunhofer Institute for Open Communication Systems, Kaiserin-Augusta-Allee 31, D-10589 Berlin, Germany; tatiana.ermakova@fokus.fraunhofer.de; 3Klinikum Mutterhaus der Borromäerinnen, Academic Teaching Hospital of Johannes Gutenberg-Universität Mainz, Feldstraße 16, D-54290 Trier, Germany; alexios.martin@mutterhaus.de

**Keywords:** T1a glottic carcinoma, transoral laser microsurgery, treatment outcome, vocal function, objective voice diagnostics, vocal extent measure (VEM)

## Abstract

Patients with unilateral vocal fold cancer (T1a) have a favorable prognosis. In addition to the oncological results of CO_2_ transoral laser microsurgery (TOLMS), voice function is among the outcome measures. Previous early glottic cancer studies have reported voice function in patients grouped into combined T stages (Tis, T1, T2) and merged cordectomy types (lesser- vs. larger-extent cordectomies). Some authors have questioned the value of objective vocal parameters. Therefore, the purpose of this exploratory prospective study was to investigate TOLMS-associated oncological and vocal outcomes in 60 T1a patients, applying the ELS protocols for cordectomy classification and voice assessment. Pre- and postoperative voice function analysis included: Vocal Extent Measure (VEM), Dysphonia Severity Index (DSI), auditory-perceptual assessment (GRB), and 9-item Voice Handicap Index (VHI-9i). Altogether, 51 subjects (43 male, eight female, mean age 65 years) completed the study. The 5-year recurrence-free, overall, and disease-specific survival rates (Kaplan–Meier method) were 71.4%, 94.4%, and 100.0%. Voice function was preserved; the objective parameter VEM (64 ± 33 vs. 83 ± 31; mean ± SD) and subjective vocal measures (G: 1.9 ± 0.7 vs. 1.3 ± 0.7; VHI-9i: 18 ± 8 vs. 9 ± 9) even improved significantly (*p* < 0.001). The VEM best reflected self-perceived voice impairment. It represents a sensitive measure of voice function for quantification of vocal performance.

## 1. Introduction

Laryngeal cancer is the most frequent malignant tumor in the head and neck area and one of the most common tumors of the respiratory tract [1,2,3]. GLOBOCAN estimates that more than 177,000 people worldwide developed laryngeal cancer in 2018, with men being affected significantly more often than women (155,000 vs. 22,000) [4]. The prognosis depends mainly on the localization, the TNM classification and the R-status, but also the differentiation and the presence of lymphangiosis carcinomatosa are relevant predictors [5,6,7]. In the glottis, squamous cell carcinomas are the most frequent type (60 to 80%) compared to other tumor sites within the larynx [8,9,10]. In early glottic cancer, carcinoma in situ (Tis) must be differentiated from T1 and T2 laryngeal cancer. Invasive T1 glottic cancer is limited to one (T1a) or both (T1b) vocal folds (VF) with normal respiratory but impaired phonatory VF mobility.

T1 and early T2 glottic carcinomas have a very good prognosis due to the early symptom of hoarseness, which usually leads to a quick diagnosis and prompt initiation of therapy. In addition, metastasis rates are low [11,12,13]. In the literature, the 5-year overall survival after therapy of early glottic cancer is reported to be in the 74–100% range [14,15]. Involvement of the anterior commissure is more likely to have higher local recurrence, lower laryngeal preservation, but no statistical difference in 5-year overall survival [16,17]. In Steiner’s landmark study of 240 patients with laryngeal cancer, early-stage carcinomas had an overall 5-year survival rate of 86.5% (disease-specific 100%), 6% local recurrences, with 99.4% larynx preservation [18]. Ledda and Puxeddu evaluated the oncologic efficacy in 103 patients with early glottic carcinoma, reporting for T1 a 5-year recurrence-free rate of 96% (local control 98%, larynx preservation 100%) [19]. Canis et al. showed in 404 pT1a patients the following 5-year Kaplan-Meier estimates: local control 86.8%, overall survival 87.8%, disease-specific survival 98.0%, recurrence-free survival 76.1%, and larynx preservation 97.3% [20]. Batra et al. presented in 53 patients with Tis and T1 comparable results: local control 86.7%, ultimate local control (with CO_2_-laser alone) 90.5%, 3-year overall survival 92.4%, 3-year disease-specific survival and larynx preservation 98.1% [21]. An analysis of 2436 transorally treated T1/T2 carcinomas showed a 5-year overall survival of 82% [22]. For disease-specific survival after T1 and T2 transoral resection, 5-year survival rates of 89–100% are reported in the literature [23]. Meta-analyses on laryngeal preservation after transoral laser resection of T1 and T2 report rates of 83–100% [24].

Early detection of laryngeal cancer can minimize surgical trauma, improve therapeutic outcome and reduce mortality [25]. It is a general consensus that the larynx should be examined laryngoscopically in all patients with hoarseness lasting more than 3 to 4 weeks [26,27]. Videolaryngostroboscopy (VLS) can indicate invasive tissue growth by eliminated mucosal wave propagation and reduced or absent phonatory VF mobility [28,29]. Electronic chromoendoscopy can improve the recognition of tumor margins [30]. A recording of connected speech to document the impaired vocal function is considered a minimum requirement for functional assessment [31]. Small glottal findings suspected of malignancy such as precursor lesions, Tis, and T1a carcinomas, can be completely removed during diagnostic microlaryngoscopy to confirm the diagnosis by excision biopsy [32,33]. Apart from the health status, the quality of life in patients with T1 glottic cancer depends mainly on the voice quality and thus on the extent of the resected VF tissue [34,35,36]. Surgical therapy is preferred [37,38]; primary radiotherapy, however, can also be used as a conservative VF preserving procedure [39,40].

Transoral CO_2_-laser microsurgery (TOLMS) was introduced by Strong and Jako for the therapy of early laryngeal cancer in the 1970s [41], and Steiner gave further impetus in the propagation of this technique [18,42]. Today, TOLMS is established for the treatment of early glottic carcinoma with highly satisfying oncological and functional outcomes (e.g., [20,43,44]). However, many studies predominately focus on oncological results and not on functional outcomes. As the vocal outcome depends on the amount of removed tissue, the consistent classification of endoscopic cordectomies of the European Laryngological Society (ELS) allows interpretation of postoperative results with regard to the surgical strategy and comparison between different surgical centers [45]. The main objective of this exploratory study was to examine in detail the vocal outcome in patients with T1a glottic cancer. The hypothesis was that voice function can be preserved after TOLMS. Therefore, we planned to explore the pre- and postoperative vocal function using specific subjective and objective parameters including the vocal extent measure (VEM) based on the voice range profile (VRP) [46].

## 2. Materials and Methods

### 2.1. Study Design and Patients

Patients diagnosed with suspected T1a glottic carcinoma underwent direct microlaryngoscopy in general anaesthesia with TOLMS in a prospective study. Clinical examination and data acquisition took place at the initial pre-therapeutic visit, during operation, and at regular follow-ups postoperatively. The voice was examined the day before TOLMS and 3 months after in-sano resection and completed wound healing. Study participants were patients presenting with hoarseness at the Department of Audiology and Phoniatrics, Charité–University Medicine Berlin, Germany. Altogether, 60 consecutive patients were recruited between June 2009 and October 2019. Selection criteria comprised histologically confirmed pT1a cN0 cM0 glottic carcinoma, complete treatment documentation, and informed consent. Patients with Tis, T1b and T2 glottic cancer were not included in this investigation.

### 2.2. Surgical Procedure and Postoperative Regimen

Microlaryngoscopy was conducted via the operating microscope type OPMI Sensera (Zeiss, Jena, Germany) and the Kleinsasser laryngoscope suspension system (Storz, Tuttlingen, Germany). TOLMS was performed with the AcuPulse 30W/40 ST CO_2_-laser system (Lumenis, Yokneam, Israel) using the following parameters: output power 2 to 5 watt, super pulse mode, continuous wave, spot size 200 µm, focal length 400 mm. Conventional intraoperative safety precautions were respected (patient covering with moist cloths, safety goggles, laser-resistant endotracheal tube, ventilation with oxygen concentration below 40%). After inspection and palpation under the microscope, saline containing epinephrine (1 mg/mL; 10 gtt. in 10 mL NaCl) was injected into the VF. As a result, stretching the epithelium allowed to assess the fixation of the lesion to deeper structures. The saline also protected the healthy surrounding VF tissue from thermal damage. Laser incisions were made at the site where the suspicious lesions could be distinguished from normal epithelium, considering a safety margin of at least 1 mm. Depending on the pre- and intraoperative findings, cordectomy was conducted. After having removed the suspicious cancerous tissue, the surgeon classified the resection type according to the cordectomy types of the ELS [45]. Lesions within the epithelial level without fixation or signs of infiltration were superficially removed en bloc. Marginal resections were taken if the complete tumor removal was uncertain. All excision biopsies were sent for histopathological examination. The guidelines of the American Joint Committee on Cancer (AJCC) were used for tumor staging [47]. Patients with histopathologically confirmed R1 status were rescheduled for follow-up resection. All TOLMS operations were performed by 5 experienced laryngologists. After surgery, patients were monitored on the ward for 1–2 nights. Before discharge, all treated patients received vocal hygiene counseling. In the event of recurring voice impairment, they were asked to present again between regular follow-up intervals. Postoperative voice rest was not recommended.

### 2.3. Examination Instruments and Criteria

The analysis of treatment outcome was based on postoperative histopathological findings, pre- and postoperative VLS, and voice function diagnostics. Digital 2D or 3D VLS was carried out via rigid transoral or flexible transnasal endoscopes with integrated microphones (XION GmbH, Berlin, Germany) [28,48]. According to the ELS protocol, voice function diagnostics consisted of established subjective (i.e., auditory-perceptual assessment, self-evaluation of voice) and objective procedures (i.e., VRP measurement, acoustic-aerodynamic analysis) [49,50,51]. Objective procedures quantify the investigated aspects of vocal function in an apparatus-based and neutral manner. Subjective tests describe the individual self-perceived vocal impairment from the examined person’s point of view as well as auditory-perceptual assessments from the examiner’s viewpoint.

Auditory-perceptual assessment of the recorded voice samples was conducted using the GRB system [31]. The perceived overall grade of hoarseness (G), roughness (R), and breathiness (B) were independently rated on a scale from 0 to 3 (0 = not existing, 1 = mild, 2 = moderate, 3 = severe) by two senior phoniatricians. From each audio recording the mean score of both GRB evaluations served for further analysis.

Subjective self-assessment of voice was obtained using the 9-item Voice Handicap Index (VHI-9i) including 9 questions rated on a scale from 0 to 4 (0 = never, 1 = almost never, 2 = sometimes, 3 = almost always, 4 = always) [52]. The VHI-9i reflects the functional, physical and emotional impact of the voice disorder on the patient’s quality of life. Additionally, an estimation of the self-perceived overall vocal impairment (VHIs) at the time of questioning was scored between 0 and 3 (0 = normal, 1 = mild, 2 = moderate, 3 = severe).

VRP measurements and acoustic-aerodynamic analyses were performed with the DiVAS software (XION GmbH) to obtain objective quantitative data of the speaking and singing voice. The following parameters were collected: soft phonation threshold, highest and lowest pitch, maximum phonation time (MPT), jitter, dysphonia severity index (DSI) [53], and VEM [46]. The VEM is the logarithmised product of the area of the VRP (A_VRP_) and the quotient of the circumference of a circle with the same area and the actual VRP circumference (P_VRP_), supplemented by the addition of a coefficient (50) and an offset (−200). The mathematical formula is:(1)$$VEM=50\ln\left(A_{VRP}\frac{2\pi \;\sqrt{\frac{A_{VRP}}{\pi }}}{P_{VRP}}\right)-200$$

The VEM quantifies the patient’s dynamic performance and the frequency range as documented in the VRP. It expresses the vocal capacity as an interval-scaled value, mostly between 0 and 120. A high vocal capacity is characterized by a high VEM; conversely, a small VRP results in a small VEM.

## 3. Data Analysis

Descriptive statistics were used to describe the quantitative features of all pre- and postoperative parameters and their changes. As graphical techniques to display the data, we chose histograms and violin plots, i.e., box plots with kernel density plots rotated and surrounding them on each side. Being suitable for both continuous and ordinal variables, Spearman’s rank-order correlation (r_s_) was used to investigate the strength and direction of association between the pre- und postoperatively measured characteristics and their differences. Wilcoxon signed-rank test was used to test whether vocal function parameters significantly improved as the result of TOLMS. Mean values and 95% confidence intervals for these changes were calculated. The impact of patient-related, tumor-related, and treatment-related factors on disease control and survival was analyzed using the Kaplan–Meier method. All statistical tests and graphics were done using R version 4.0.1 (GNU project, Free Software Foundation, Boston, MA, USA). The level of significance was set at α = 0.05. Due to the exploratory nature of the study no adjustment for multiple testing was performed. To show different significance levels, the following abbreviations were used: * = 5%; ** = 1%; *** = 0.1%.

## 4. Results

### 4.1. Sample Description and Preoperative Assessment

From 60 patients initially recruited with histopathologically confirmed diagnosis of pT1a, six subjects (10.0%) were lost to follow-up and three subjects (5.0%) had to be excluded due to incomplete treatment documentation. In the remaining 51 patients, all diagnostic tests and therapeutic procedures were carried out as planned. The total sample consisted of 43 men and 8 woman, with a mean age of 65 years (range 31–84). At the time of intervention, women were on average 16 years younger than men (52 ± 14 vs. 68 ± 10, mean ± SD, *p* < 0.01). Regarding medical history, 39 subjects (76.5%) gave information about current or past tobacco abuse, with 12 subjects (23.5%) having smoked rarely or not at all. While 15.7% of the patients (8/51) never drank alcohol, 62.7% (32/51) reported regular and 21.6% (11/51) daily consumption of alcohol. Relevant preoperative patient characteristics within the examined cohort are shown in Table 1 (left side).

VLS revealed an almost equal distribution of tumor growth on both VF (28 right, 23 left). The lesions appeared flat and hyperkeratotic in 20/51 (39.2%), exophytic in 29/51 (56.9%), and ulcerating in 2/51 (3.9%) subjects. Concerning macroscopic assessment of tumor size at initial presentation, 51.0% of the patients (26/51) showed involvement of the entire VF, while in 27.4% (14/51) two-thirds and in 21.6% (11/51) one-third of the VF were affected. During phonation, phonatory VF mobility was reduced or absent on the affected tumor side in all subjects. Additionally, patients with bulged VF due to exophytic tissue growth displayed highly impaired glottal closure.

Subjective auditory-perceptual evaluation of patient’s voices was categorized preoperatively with a mean of G2 R2 B1 (range 0–3). The VHI-9i had an average score of 18 ± 8, corresponding to moderate self-assessed patient complaints. The objective acoustic and aerodynamic parameters also indicated moderate impairment (e.g., VEM 64 ± 33; DSI 1.2 ± 2.4; MPT 13 ± 6 s). Correlation analysis performed on preoperative values showed that both VEM and DSI correlated with VHI-9i (r_s_ = −0.62*** and r_s_ = −0.29*, respectively), G (r_s_ = −0.42** and r_s_ = −0.34*), R (r_s_ = −0.41** and r_s_ = −0.37**), B (r_s_ = −0.47*** and r_s_ = −0.30*), and with each other (r_s_ = 0.51***).

### 4.2. Postoperative Assessment

Via TOLMS, 24 patients received subepithelial cordectomy (type I; 47.1%), 18 patients subligamental cordectomy (type II; 35.3%), and nine patients transmuscular cordectomy (type III; 17.6%). According to histopathology, the diagnosis confirmed in all subjects squamous cell carcinoma limited to one VF (pT1a). The grading classification revealed in most patients moderately differentiated tissue (G2; 66.7%), less frequent well differentiated (G1; 29.4%) and seldom poorly differentiated tissue (G3; 3.9%). Through primary operation, the pT1a was completely excised (R0 status) in 29 patients (56.9%). Following the piecemeal strategy, a second excision was necessary in 22 subjects (43.1%), as a residuum could not be ruled out (close tumor margin vs. R1 status). Of these 22 subjects with suspicious findings, 17 patients (77.3%) had no visual or histopathological malignant residue in the scheduled control TOLMS. Among the remaining five patients, the follow-up resections revealed residual invasive tumor in three patients (13.7%), Tis in one patient (4.5%), and a precursor lesion (squamous intraepithelial neoplasia SIN III) in the other patient (4.5%). All these lesions were completely excised during the second TOLMS.

The operative procedures were conducted without complications. Postoperatively, no patient complained about swallowing dysfunction. VLS check-ups showed fibrin formation on the wound surfaces followed by formation of scar tissue during healing. While extensive tumor growth was associated with larger glottal defects after removal, in smaller superficial findings treated via type I cordectomy a stable epithelium regenerated on the preserved lamina propria without relevant defects or scarring. In some patients, the scarred VF developed after about 6 months a restored phonatory mobility. Figure 1 gives an impression of pre- and postoperative VLS findings with videostrobokymographic illustration of VF oscillations.

Within the mean postoperative observation period of 45 ± 26 months (median: 41 months), 10 patients (19.6%) suffered from a local recurrence (1× Tis, 7× rpT1a, 1× rpT1b, 1× cT3) with an average tumor-free interval of 15 months (median 10 months). Eight of these subjects had only one recurrence within the follow-up period. Among the remaining two, further recurrences occurred: one patient with the initial diagnosis of pT1a (G3) suffered from two recurrences of rpT1a after 17 and 80 months. The other subject with the initial diagnosis of pT1a (G2) had altogether four recurrences; after 13 (rpT1a), 27 (rpT2), 44 (rT3), and 92 months (rpT4a). During follow-up, a secondary glottic pT1a on the contralateral VF was detected in two patients after an interval of 1 and 3 years after removal of the primary tumor, respectively. All recurrent and secondary laryngeal carcinomas were successfully treated: Tis, T1 and T2 via secondary TOLMS, both T3 recurrences via radio-chemotherapy, and the T4 recurrence via total laryngectomy. One subject died due to a secondary pancreas carcinoma, another one died intercurrently. The 5-year recurrence-free, overall, and disease-specific survival rates (Kaplan–Meier method) were 71.4%, 94.4%, and 100.0% (Figure 2). Relevant postoperative and oncological patient characteristics are shown in Table 1 (right side).

Three months after TOLMS, vocal function improved considerably compared to the preoperative measurements (Table 2). With respect to auditory-perceptual GRB evaluation, the pre- vs. post-therapeutical comparison revealed that the voices were less hoarse (1.9 ± 0.7 vs. 1.3 ± 0.7), rough (1.8 ± 0.7 vs. 1.2 ± 0.7), and breathy (1.0 ± 0.6 vs. 0.6 ± 0.6). The subjective vocal self-assessment via VHI-9i questionnaire demonstrated a mean reduction from 18 ± 8 to 9 ± 9 points. The VHIs criterion indicated a change from moderately (2 ± 1) to mildly disturbed voices (1 ± 1). The improvements regarding all these subjective parameters were found significant at the 0.1% level (*p* < 0.001). The subjective vocal parameters both pre- and postoperatively are displayed by histograms in Figure 3.

Regarding objective measures, the VEM improved significantly in the total cohort (from 64 ± 33 to 83 ± 31; *p* < 0.001), in both genders (males *p* < 0.01; females *p* < 0.05) and all cordectomy types (*p* < 0.05). In contrast, the decrease of jitter (0.9 ± 1.1 to 0.6 ± 0.4) and the increase of DSI (1.2 ± 2.4 to 1.5 ± 2.3) did not reach the level of significance in the total group, only in females (*p* < 0.05) and cordectomy type III (*p* < 0.05). VEM and DSI correlated significantly with each other also postoperatively (r_s_ = 0.62***). The VEM showed a significant negative correlation with VHI-9i (r_s_ = −0.29*) but not with age (r_s_ = −0.18), while the DSI correlated significantly with age (r_s_ = −0.39**) but not with VHI−9i (r_s_ = −0.11). Selected objective parameters before and after pT1a removal are graphically displayed via boxplots in Figure 4 with regard to the total cohort and cordectomy type.

To provide insights into the magnitude of changes induced by TOLMS, Table 2 also presents the mean differences (and 95% confidence intervals) between pre- and post-therapeutic values. As a result, the numeric outcome of all subjective and objective parameters was larger in women compared to men. Similarly, the improvement of these parameters in cordectomy type III was higher compared to the other cordectomy types.

## 5. Discussion

Given the established favorable oncological results of CO_2_-TOLMS in T1a glottic carcinoma, functional aspects should be another treatment objective. We successfully examined the oncological and functional outcomes after TOLMS in pT1a patients, focusing on the evaluation of voice with subjective and objective parameters. Our T1a cohort is consistent with the literature in terms of patient characteristics, treatment methods, and oncological results (see Table 1, Figure 2). Therefore, a closer look at our vocal outcomes is warranted compared to the results of previous investigations.

Many studies were conducted to compare TOLMS with radiotherapy in patients with early glottic cancer [54,55,56]. The vocal outcomes were either superior in radiotherapy [57,58] or in TOLMS [59,60], or they did not show relevant differences between both treatment groups [61,62,63,64]. In general, pre-therapeutic voice data was often not collected [57,58,59,61,63,64,65,66,67,68,69]. In these investigations, it is impossible to relate the postoperative voice function to the pretherapeutic baseline. Some studies evaluated vocal function before and after TOLMS according to the cordectomy type [70,71,72,73,74]. Mainly, voice quality differed depending on the amount of tissue resected: vocal outcomes after lesser-extent cordectomies (ELS type I, II) were superior compared to larger-extent cordectomies. However, a multidimensional, detailed pre- and post-therapeutic documentation and evaluation of voice was only carried out in a few studies [62,70,71,74,75]. To compare the vocal outcomes after TOLMS, Table 3 summarizes the main results of previous investigations including the number of T1a patients treated and the parameters used for evaluation.

The comparability of published studies is limited due to the lack of standardization regarding (1) vocal outcome assessment (different parameters, follow up), (2) patient selection (e.g., all early glottic cancer patients, low number of T1a), as well as (3) inclusion and treatment criteria (e.g., combined T stages and cordectomy types).

The usefulness of objective acoustic measures has been questioned. Some studies indicated that TOLMS results in an increase of F0, jitter, shimmer, and a moderate decrease of MPT in extended cordectomies when compared with healthy controls (e.g., [79]). Other studies found either a TOLMS-associated improvement [74,75,77], or no relevant changes throughout the postoperative course [70,78]. In our investigation, the patients revealed in all objective and subjective parameters postoperative changes. Similar to the literature, subjective parameters improved significantly [71,72,77,79]: GRB, VHI-9i and VHIs substantially improved in our total cohort, both genders, and in each cordectomy group. Among objective measures, the MPT showed non-specific, undirected changes without any significance. This is in concordance with the results of Hamzany et al., confirming that aerodynamic parameters seem to be less suitable for outcome assessment in T1a glottic carcinoma [70]. Regarding acoustic parameters, VEM seems to be very well suited to assess the resulting voice function after T1a excision compared to other objective acoustic parameters, as only this measure responded significantly in the total cohort and in all subgroups. Among cordectomy types, the larger the resections, the greater the postoperative subjective numerical benefit (Table 2). Similarly, the improvement of acoustic parameters in cordectomy type III was bigger compared to the other cordectomy types. This is related to the fact that larger tumors are associated with more severe voice impairment preoperatively. In contrast, better voice function in smaller tumors results in less postoperative numerical benefit, even if the final voice outcome is better. The relevant differences in the cordectomy groups (types I–III) suggest that pooling these types, as in previous studies of the literature, does not seem appropriate. Although all subjective and objective improvements were larger in women than men, we cannot draw general conclusions due to our limited number of female patients.

While the VEM is not yet widely applied in voice diagnostics, the multidimensional DSI represents an established parameter of instrumental voice evaluation based on a weighted combination of highest possible frequency, lowest intensity, MPT and jitter [53]. Former investigations showed that the DSI might be influenced by using different registration programs, as well as by age or gender [80,81]. These age and gender effects were also confirmed in our study. The DSI appears susceptible to extreme measures (e.g., highest frequency, lowest intensity), which are likely to be influenced by age or gender. In contrast, the VEM, calculated from area and shape of the VRP, is less affected by the above-mentioned extreme measures. Since VEM correlated highly significantly with DSI, both measurements can be seen as related and comparable parameters. Part of their shared variance could be accountable to age, although the linear relationship with age is considerably weaker for the VEM compared to the DSI. However, the VEM as a positive criterion characterizes the vocal abilities and enables a classification of voice performance, while the DSI as a negative criterion particularly describes the severity of dysphonia [80,82]. Among both parameters, the VEM better reflected the subjective vocal impairments. However, DSI, VEM, VHI, and GRB represent different aspects of the voice: They are complementary in objective and subjective evaluation of voice quality, vocal performance, or perceived vocal handicap.

Depending on preoperative T1a tumor characteristics, individual postoperative voice function might be better, similar, or slightly reduced. In general, objective and subjective voice quality improved during long-term postoperative follow-up. This is in line with the results of previous investigations [70,83]. Although voice diagnostics according to ELS protocol is more time-consuming, we consider this effort justified for evidence-based therapy and necessary for documentation of voice preservation. To preserve voice function, the intraoperative laser power should be selected as low as possible to avoid thermal damage in the surrounding healthy tissue. In addition, focused excision achieves better vocal outcomes than defocused vaporization [62]. The application of the KTP laser may be able to offer improved voice preservation with similar oncological control compared to CO_2_-TOLMS [76,77]. The focus on voice preservation may increase the number of interventions in cases with histologically questionable tumor margins [84,85]. Our experience confirms the literature, that re-operation can sometimes be avoided by close monitoring of local control using VLS [44,66].

### Study Strengths and Limitations

Our study is characterized by the application of multidimensional voice evaluation, extended by the objective VEM. Further strengths comprise cohort homogeneity restricted to T1a instead of all early glottic cancer patients, and evaluation of specific cordectomy types in a sufficient number of patients rather than generalization or grouping into lesser- vs. larger-extent cordectomies. Applying the ELS protocols both for cordectomy classification and multidimensional voice evaluation enables a systematic comparison of our results with the outcomes of future studies.

Some limitations must be considered before drawing general conclusions. First, our results are investigations of a mono-centre study. To prevent centre bias, multicentre trials with a larger number of subjects are needed. Second, females are underrepresented in our study; thus, there may be participation bias. With a limited number of female patients, general gender-specific conclusions cannot be drawn. Our study sample reflects the well-known prevalence of laryngeal cancer in male patients, though. Third, a more precise preoperative assessment of the exact extent of the pathology would be useful. The importance of tumor size and shape should not be underestimated regarding voice function. The histopathologically determined tumor extent does not replace this information, because resections via TOLMS are not always performed en bloc and may lead to thermal tissue artefacts (e.g., shrinkage, coagulation, vaporization). Fourth, there were differences regarding the individual amount of interventions as well as rehabilitation strategies. Voice therapy could influence the vocal outcome in operated patients. Having neglected this may also result in a performance bias. Lastly, some factors influencing the VRP registration have to be considered. One limitation is the fact that in aphonic patients no perimeter of the VRP can be measured. However, in our study no T1a patient suffered from aphonia. Other factors comprise the routine of the examiner, motivation of the patients, and varying quantities of registered tones. Most of these influential factors are of minor importance in our investigation because all VRPs were recorded by one experienced examiner under practically equal conditions. Since precise VEM calculation is based on the actual VRP shape and circumference, future multicenter studies should be standardized by defining the number of registered tones per interval.

## 6. Conclusions

TOLMS has been proven to be an established and safe standard oncologic therapy for T1a glottic carcinoma with satisfactory preservation of vocal function both subjectively and objectively. Among objective voice parameters, the VEM seems to best reflect self-perceived subjective voice impairment showing significant changes after T1a treatment that incorporates phonosurgical principles. It represents a sensitive, positive measure of voice function, as well as an understandable and easy-to-use parameter for quantifying vocal performance as documented in the VRP. Therefore, it is reasonable to include the VEM as a diagnostic addition to the established voice measures of the ELS protocol.

## Figures and Tables

**Figure 1 jcm-10-01250-f001:**
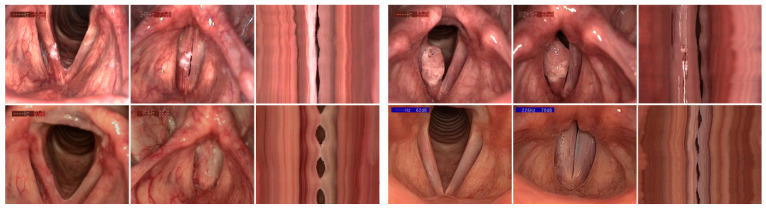
Videolaryngostroboscopic pictures and videostrobokymographic illustration of vocal fold anatomy and function, preoperative (**upper row**) vs. postoperative (**lower row**). Example A (**left side**): 45-year-old male professional theater actor with a flat hyperkeratotic lesion of the right vocal fold. Example B (**right side**): 32-year-old female medical doctor with an exophytic tumor of the right vocal fold. Findings three months postoperatively show: pT1a completely removed, healing process finished, vocal folds with straight margin, complete glottal closure, and restored phonatory mobility (A: normalized, regular and symmetric oscillations; B: oscillations with scarring-related reduced amplitude and phase shift).

**Figure 2 jcm-10-01250-f002:**
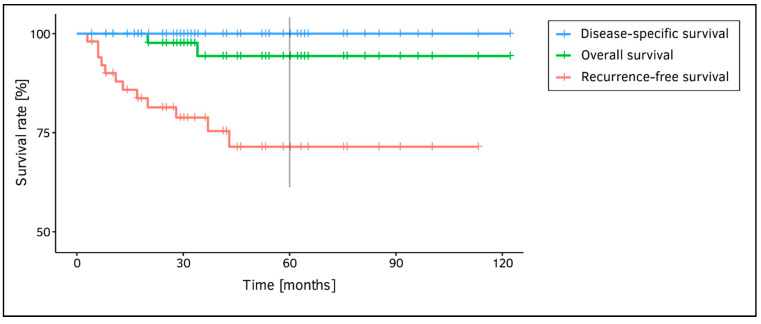
Five-year Kaplan–Meier estimates for recurrence-free survival, overall survival, and disease-specific survival.

**Figure 3 jcm-10-01250-f003:**
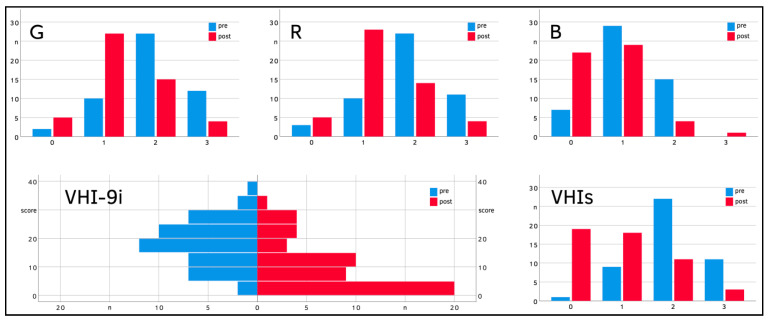
Subjective vocal parameters before and after pT1a removal. Upper row: Comparison of pre- and postoperative voice parameters according to the GRB-classification. Lower row: Comparison of pre- and postoperative VHI-9i and VHIs scores.

**Figure 4 jcm-10-01250-f004:**
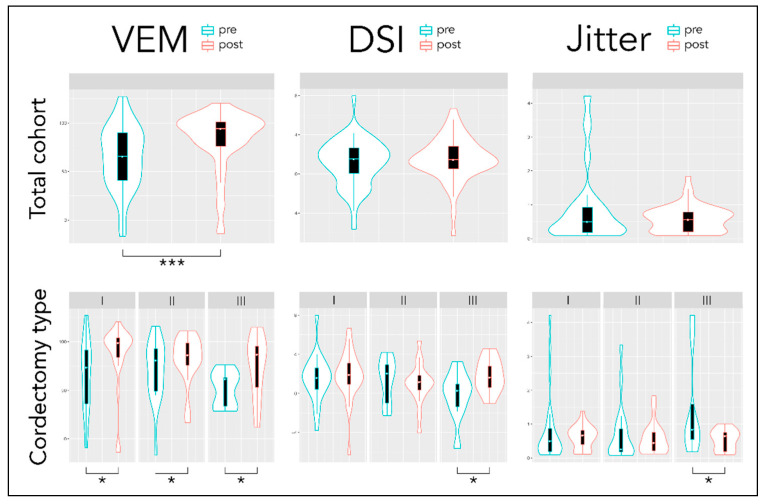
Objective acoustic parameters VEM, DSI, and jitter before and after pT1a removal concerning the total cohort and cordectomy types. Data are compared pre- vs. postoperatively via violin plots, i.e., box plots with kernel density plots rotated and surrounding them on each side. The boxplots display the median, quartiles, and the range of values covered by the data. The density curves display the full distribution of the data including any outliers. The level of significance is indicated as follows: * significant at *p* < 0.05; ** significant at *p* < 0.01; *** significant at *p* < 0.001 (Wilcoxon signed-rank test).

**Table 1 jcm-10-01250-t001:** Patient characteristics (*n* = 51) before TOLMS (left) and after TOLMS (right). Unless otherwise specified, data expressed as number of patients and percentage of group.

	Number	%			Number	%
Gender male female	43 8	84.3% 15.7%		Initial cordectomy (via TOLMS) type I (subepithelial) type II (subligamental) type III (transmuscular)	24 18 9	47.1% 35.3% 17.6%
Age (in years; mean ± SD)	65 ± 12	-		Grading of pT1a G1 (well differentiated) G2 (moderately differentiated) G3 (poorly differentiated)	15 34 2	29.4% 66.7% 3.9%
Occurrence of pT1a left vocal fold right vocal fold	23 28	45.1% 54.9%		Follow-up (in months; mean ± SD)	45 ± 26	-
Vocal fold involvement anterior third middle third posterior third anterior and middle third middle and posterior third entire length	3 7 1 7 7 26	5.9% 13.7% 2.0% 13.7% 13.7% 51.0%		Treatment response local disease control local disease recurrence contralateral secondary pT1a ultimate local disease control with TOLMS alone) larynx preservation	41 10 2 49 50	80.4% 19.6% 3.9% 96.1% 98.0%
Appearance of pT1a hyperkeratotic exophytic ulcerating	20 29 2	39.2% 56.9% 3.9%		Survival disease-specific overall recurrence-free	51 49 39	100.0% 96.1% 76.5%

**Table 2 jcm-10-01250-t002:** Pre- and posttherapeutic parameters of vocal function in all patients and all cordectomy types (mean ± SD), their mean therapeutic differences (Diff) and 95% confidence intervals (CI) for changes in vocal measures three months after pT1a removal.

Vocal Measure		Total Group (*n* = 51)	Type I Cordectomy (*n* = 24)	Type II Cordectomy (*n* = 18)	Type III Cordectomy (*n* = 9)
VEM	Pre Post	64.4 ± 32.7 82.8 ± 30.5	65.4 ± 36.9 86.7 ± 33.5	70.3 ± 31.7 81.9 ± 25.4	51.0 ± 18.4 74.1 ± 33.2
	Diff (CI)	18.4 (9.0; 29.8) ***	21.3 (5.1; 37.6) *	11.6 (−3.2; 32.6) *	23.1 (−5.7; 52.0) *
DSI	Pre Post	1.2 ± 2.4 1.5 ± 2.3	1.5 ± 2.4 1.8 ± 2.6	1.4 ± 2.3 1.0 ± 2.1	−0.2 ± 2.6 1.8 ± 1.8
	Diff (CI)	0.3 (−0.2; 1.3)	0.3 (−0.5; 1.9)	−0.4 (−1.4; 0.6)	2.0 (0.1; 3.9) *
Jitter (%)	Pre Post	0.9 ± 1.1 0.6 ± 0.4	0.8 ± 1.1 0.6 ± 0.3	0.7 ± 0.9 0.6 ± 0.5	1.5 ± 1.6 0.5 ± 0.3
	Diff (CI)	−0.3 (−0.7; −0.02)	−0.2 (−0.7; 0.2)	−0.1 (−0.7; 0.3)	−1.0 (−2.0; 0.1) *
MPT (s)	Pre Post	13.3 ± 5.6 13.3 ± 6.0	14.1 ± 5.2 14.7 ± 6.3	12.3 ± 6.6 10.9 ± 5.7	13.3 ± 4.5 14.6 ± 4.5
	Diff (CI)	−0.01 (−1.9; 1.9)	0.6 (−2.4; 3.6)	−1.4 (−4.6; 1.7)	1.3 (−3.6; 6.0)
VHI−9i	Pre Post	17.7 ± 8.1 9.3 ± 8.8	16.6 ± 8.3 10.5 ± 9.0	17.1 ± 7.1 7.7 ± 8.7	22.1 ± 9.1 9.2 ± 8.8
	Diff (CI)	−8.4 (−10.9; −5.6) ***	−6.1 (−10.5; −2.1) **	−9.4 (−13.1; −4.9) **	−12.9 (−20.4; −4.3) *
VHIs	Pre Post	2.0 ± 0.7 1.0 ± 0.9	1.9 ± 0.9 1.0 ± 1.0	1.9 ± 0.6 0.8 ± 0.9	2.4 ± 0.5 1.0 ± 0.9
	Diff (CI)	−1.0 (−1.4; −0.8) ***	−0.9 (−1.3; −0.6) ***	−1.1 (−1.7; −0.7) ***	−1.4 (−2.2; −0.6) *
G	Pre Post	1.9 ± 0.7 1.3 ± 0.7	1.5 ± 0.8 1.0 ± 0.8	2.2 ± 0.4 1.5 ± 0.6	2.2 ± 0.7 1.4 ± 0.6
	Diff (CI)	−0.6 (−0.8; −0.4) ***	−0.5 (−0.8; −0.2) **	−0.7 (−0.9; −0.4) **	−0.8 (−1.2; −0.2) *
R	Pre Post	1.8 ± 0.7 1.2 ± 0.7	1.5 ± 0.8 1.0 ± 0.8	2.1 ± 0.5 1.5 ± 0.6	2.0 ± 0.8 1.3 ± 0.6
	Diff (CI)	−0.6 (−0.8; −0.4) ***	−0.5 (−0.8; −0.2) **	−0.6 (−0.9; −0.3) **	−0.7 (−1.2; −0.1) *
B	Pre Post	1.0 ± 0.6 0.6 ± 0.6	0.8 ± 0.7 0.4 ± 0.6	1.2 ± 0.4 0.9 ± 0.5	1.4 ± 0.4 0.9 ± 0.7
	Diff (CI)	−0.4 (−0.6; −0.2) ***	−0.4 (−0.7; −0.1) **	−0.3 (−0.6; −0.1) **	−0.5 (−1.1; 0.1) *

B: breathiness; DSI: dysphonia severity index; G: (overall) grade of hoarseness; MPT: maximum phonation time; R: roughness; VEM: vocal extent measure; VHI-9i: 9-item voice handicap index, VHIs: self-perceived overall vocal impairment. The level of significance is indicated as follows: * significant at *p* < 0.05; ** significant at *p* < 0.01; *** significant at *p* < 0.001 (Wilcoxon signed-rank test).

**Table 3 jcm-10-01250-t003:** Published vocal outcomes for T1a glottic cancer treated with TOLMS, taken from representative studies (last 14 years, *n* > 10 T1a patients operated via TOLMS).

Study	Numbers	Parameters for Evaluation of Vocal Function			Vocal Outcome after Transoral Lasermicrosurgery (TOLMS)
		Clinician-Rated Assessment (Subjective)	Patient’s Self-Assessment (Subjective)	Acoustic-Aerodynamic Evaluation (Objective)	
Hamzany et al. (2021) [70]	27 T1a	GRB	VHI	F0, jitter, shimmer, NHR, MPT	significant subjective improvement, no objective improvement
Strieth et al. (2019) [76]	14 T1a	–	VHI	–	improved voice preservation by KTP-TOLMS (lower VHI scores) compared to CO_2_-TOLMS (higher VHI scores)
Gandhi et al. (2018) [59]	40 T1a + b (N/S)	GRBAS	VHI	F0, jitter, shimmer, SPI, NHR	excellent vocal outcome (G 0.63, VHI 13); no pretherapeutic data
Hong et al. (2018) [61]	14 T1a + b (N/S)	GRBAS	–	F0, jitter, shimmer, NHR	GRB with mild dysphonia, Jitter 2.37%; no pretherapeutic data
Lee et al. (2016) [71]	50 T1a	GRBAS	VHI	F0, jitter, shimmer, NHR, voice intensity, MPT	G significantly improved; voice quality improved over time in limited ELS resections (I-II) but not in extended cordectomies (III-V)
Fink et al. (2016) [72]	38 T1a	VAS (0–100)	VHI	–	similar or improved voice in limited ELS resections (I-III), VHI improved significantly (VAS n.s.); poorer outcomes in extended resections
Kono et al. (2016) [62]	64 T1a	GRBAS	VHI, V-RQOL	F0, jitter, shimmer, NHR, MPT	mild to moderate impairment (GRB, VHI, jitter), better improvement over time in focused excision compared to defocused vaporization
Berania et al. (2015) [65]	18 T1a	PSS-H&N	VHI-10	–	favorable functional outcomes (40% mild voice handicap, VHI-10 > 11); no pretherapeutic data
Bertino et al. (2015) [66]	135 T1a	degree of dysphonia (acc. Ricci Maccarini)	–	F0, HNR	mild to slight dysphonia in limited ELS resections (I-II), moderate to severe dysphonia in extended resections (III-V); no pretherapeutic data
Laoufi et al. (2014) [57]	44 T1a	–	VHI, EORTC QLQ-HN35	–	VHI score mild to moderate impaired (mean 29); no pretherapeutic data
Friedman et al. (2013) [77]	57 T1a	–	V-RQOL	F0, jitter, shimmer, NHR, max. SPL range, max. F0 range, SPL divided by subglottic pressure	significant improvement of subjective (V-RQOL) and most objective (acoustic, aerodynamic) measures
Tomifuji et al. (2013) [73]	33 T1a	GRBAS	VHI	jitter, shimmer, HNR, MPT, MFR	voice quality differs according to the type of cordectomy; no pretherapeutic data
van Gogh et al. (2012) [60]	67 T1a	–	–	F0, jitter, shimmer, NNE	quick voice outcome recovery apart from F0 (remains higher pitched), no significant long-term voice changes
Bajaj et al. (2011) [67]	14 T1a + b (N/S)	GRBAS	VoiSS, UW-QoL	F0, F0 irregularity, CQ range, CQ irregularity	preservation of acceptable vocal function (GRB mild to moderate impaired, low VoiSS score); no pretherapeutic data
Keilmann et al. (2011) [68]	11 T1a	RBH	VHI-12	F0, jitter, shimmer, MPT, GHD, VRP	discrepancy over time (VHI deteriorated; RBH and objective measures improved); no pretherapeutic data
Lester et al. (2011) [78]	19 T1a + b (N/S)	–	ordinal scale (1–5)	F0, jitter, shimmer, MPT	objective acoustic measures showed no significant changes; deterioration of MPT (13s to 12s) and subjective rating score (3 to 2)
Motta et al. (2008) [69]	49 T1a	–	–	MPT HNR, average voice intensity	outcomes vary in relation to the main site of the pseudo-glottis, vocal compensation without normal voice quality; no pretherapeutic data
Núñez Batalla et al. (2008) [63]	19 T1a	GRBAS	VHI	F0, jitter, shimmer, NNE, MPT	mild to moderate impairment (GRBAS, VHI); no pretherapeutic data
Sjögren et al. (2008) [64]	18 T1a	GRBAS	VHI	F0, jitter, shimmer, intensity, MPT, VC, phonation quotient	mild to moderate voice dysfunction (G, B, VHI) in ca. half of patients; no pretherapeutic data
Vilaseca et al. (2008) [79]	35 T1a	GRBAS	ordinal scale (1–3)	F0, jitter, shimmer, NHR, vocal range, MPT	self-assessed improvement; compared with healthy controls: increase of F0, jitter, shimmer (MPT decrease in extended resections); no pretherapeutic data
Roh et al. (2007) [75]	50 T1a	GRBAS	VHI, EORTC QLQ-HN35	F0, jitter, shimmer, HNR, MPT, average airflow	improved vocal outcomes, significant in type I and II cordectomies (VHI, G, jitter, shimmer, HNR)

**Legend:** CQ—closed quotient, EORTC QLQ-HN35—European Organization for Research and Treatment of Cancer Head and Neck Quality of Life questionnaire; F0—fundamental frequency; GHD—Goettinger Hoarseness Diagram; GRBAS—overall Grade, Roughness, Breathyness, Asthenia, Strain; NHR—harmonics-to-noise ratio; KTP—Potassium titanyl phosphate; MFR—mean flow rate; MPT—maximum phonation time; NHR—noise-to-harmonic ratio; NNE—normalized noise energy; N/S—not specified; PSS-H&N—performance status scale for head & neck cancer patients; RBH—Roughness, Breathyness, (overall grade of) Hoarseness; SNR—signal to-noise ratio; SPI—soft phonation index; SPL—sound pressure level; UW-QoL—University of Washington Quality of Life questionnaire; VAS—visual analogue scale; VC—vital capacity; VHI—voice handicap index; VHI-10—10-item VHI; VHI-12—12-item VHI; VoiSS—voice symptom scale; VRP—voice range profile; V-RQOL—Voice-Related Quality-of-Life survey.

## Data Availability

All data of the study are available in the Department of Audiology and Phoniatrics, Charité–Universitätsmedizin Berlin, Berlin, Germany.

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
