# Peer review of "T1a Glottic Cancer: Advances in Vocal Outcome Assessment after Transoral CO2-Laser Microsurgery Using the VEM"

_jcm, 2021, doi:10.3390/jcm10061250_

Round 1
Reviewer 1 Report
General Comments
(1) The study used „the new Vocal Extent Measure (VEM)“ as an outcome measure. References indicate the VEM was only recently developed and first published in 2018 (Caffier, et al). Yet it is said that this study was conducted prospectively from 2009 and 2019 (line 87). I wonder how a prospective study can be conducted with a measure that does not yet exist.
(2) The authors speak of „objective“ and „subjective“ measures (eg. lines 74, 130, 297). It would be helpful if they provide a definition of what is meant with „subjective“ and „objective“ in this context, and which of the measures used they consider to be objective or subjective ones, respectively.
(3) „Mann-Whitney-Wilcoxon test was used to test whether vocal function parameters significantly improved as the result of TOLMS.“ – This test may not be appropriate here, as your data are dependent. Consider recalculation using paired-sample t-test, or Wilcoxon signed-rank test instead.
(4) Line 391f: „In contrast, the VEM (...) is not impaired by these interacting factors [i.e. age and gender].“ – In line 278, however, a rank correlation coefficient of -0,18 between VEM and age is reported. So the VEM is influenced by age. As the effect of gender on the VEM was not investigated in this study, you cannot rule it out.
(5) Line 393f: „Since VEM correlated highly significantly with DSI, both measurements can be seen as related and comparable parameters“. Be cautious, the correlation may be not so high as reported, as both VEM and DSI correlate negatively with age (see lines 276ff). Part of their shared variance could be accountable to age.
Considering, that the VEM and the DSI are positively correlated – how can you say that the two measures are comparable, but the VEM is a positive criterion (characterizing the vocal abilities), while the DSI is a negative criterion (describing the severity of dysphonia). In this case, their correlation should be negative.
(6) While the manuscript body is well written, the abstract is cumbersome to read. E.g. the sentence „In addition to the oncological results of CO2 transoral laser microsurgery (TOLMS) in unilateral vocal fold cancer (T1a), voice function belongs to the outcome measures.“ counts 25 words but bears little information. Likewise, the sentence „As a secondary outcome, voice function was preserved; most objective and all subjective vocal parameters even improved.“ tells the reader very little about the factual outcomes. I suggest re-writing the abstract.
Minor points:
Line 36: Who is GLOBOCAN?
Line 55: „A text recording“ – unclear what is meant here. Should read: „Recording of an audio sample to document...?“
Lines 66-68: The sentence „The consistent classification...“ does not fit in here, should be shifted to elsewhere or deleted.
Lines 75ff: Description of the VEM and its algorithm belongs into the Methods section.
Pages 8 & 9: It is a bit confusing that „VHI“ is sometimes used instead of „VHI-9i“, so that there appear to be three different questionnaires: „VHI“, „VHI-9i“ and „VHIs“. Use only and always the latter two! „VHI“ should be restricted to the 30-question version of this questionnaire.
Line 323: should read: „a good functional outcome should be another objective of the treatment.“
Line 451: Use „sensitive“ instead of „robust“ („robust“ means: unlikely being biased; this is not the point here). – What is a „positive“ measure?
Reviewer 2 Report
Title and topic: As the title mentions, the main focus should be the voice outcomes, since the literature on oncological outcomes of T1a treated by laser cordectomy is well known, and this study does not add to the existing literature.
Introduction / Discussion:
References and background regarding early glottic cancers should be enriched. For example, the impact of anterior commissure involvement should be mentioned line 39. Besides major papers a/o meta-analysis on voice outcomes in T1a glottic carcinoma treated by laser are also missing.
Introduction:
- Text recording to document the impaired vocal function is considered a minimum requirement for functional assessment [22]: by who is it recommended and how is that performed?
- "The main intention of this study was to examine in detail the vocal outcome after TOLMS in patients with T1a glottic cancers a secondary outcome measure": so is the main objective or the secondary? My opinion is that it should be the main objective, and survival should be secondary.
Methods: see comment on Table 3.
Results:
- Table 1 & 2 & 3, Figure 4, Text: with only 8 female, there is no need to separate the data in the left 4 columns (table 1 & 2) and in table 3 (columns 4 & 5), etc....
- Tables 1 & 2 could be merged into a single one presenting the cohort.
- Figure 2 is too large and not needed if oncological outcomes are a secondary objective. Table 2 is sufficient.
- Table 3. Please add in the legend the statistical tests applied here in (i) pre vs post and (ii) cordectomy types comparisons. Also, clarify in the corresponding section in the methods. If not performed yet, please use adjusted p-value for multiple tests.
- Figure 4: add visible individual dots to the box plots. Add info on stat test in the legend.
Discussion:
- The first paragraph on oncological outcomes could be shortened. However, since it is not the main objective of the study and that oncological outcomes presented here are as expected, a simple mention of that point with a range of outcomes found in the literature would be sufficient. The point here is just to say that the close consideration of your results on vocal outcomes is interesting to consider because your cohort is normal as per patient characteristics, treatment methods and outcomes.
- Vocal evaluation: the authors list many studies but do not present clearly their conclusions in a concise way. An additional table presenting the data collected in the literature and compared to the results of the present study would be very helpful and really justify the publication of this paper.
English language and style: please review minor errors in style, missing words...
Round 2
Reviewer 2 Report
General objective: You should not confuse the main objective for the patients (obviously oncological outcomes) and the main objective of your study (voice outcomes). Voice outcome needs to be the main objective of this study to make it important as compared to previous studies.
Figure 4: To show all individual dots, you need do full dot plots, not box plots with only the outlier dots visible.
